# OpenLane-V2: A Topology Reasoning Benchmark for Unified 3D HD Mapping

**Huijie Wang**[1*], **Tianyu Li**[1*], **Yang Li**[1*], **Li Chen**[1], **Chonghao Sima**[1], **Zhenbo Liu**[2],
**Bangjun Wang**[1], **Peijin Jia**[1], **Yuting Wang**[1], **Shengyin Jiang**[1], **Feng Wen**[2],
**Hang Xu**[2], **Ping Luo**[1], **Junchi Yan**[1], **Wei Zhang**[2], **Hongyang Li**[1]

[1]OpenDriveLab, Shanghai AI Lab    [2]Huawei Noah's Ark Lab

https://github.com/OpenDriveLab/OpenLane-V2

## Abstract

Accurately depicting the complex traffic scene is a vital component for autonomous vehicles to execute correct judgments. However, existing benchmarks tend to oversimplify the scene by solely focusing on lane perception tasks. Observing that human drivers rely on both lanes and traffic signals to operate their vehicles safely, we present **OpenLane-V2**, the first dataset on topology reasoning for traffic scene structure. The objective of the presented dataset is to advance research in understanding the structure of road scenes by examining the relationship between perceived entities, such as traffic elements and lanes. Leveraging existing datasets, OpenLane-V2 consists of 2,000 annotated road scenes that describe traffic elements and their correlation to the lanes. It comprises three primary sub-tasks, including the 3D lane detection inherited from OpenLane, accompanied by corresponding metrics to evaluate the model's performance. We evaluate various state-of-the-art methods, and present their quantitative and qualitative results on OpenLane-V2 to indicate future avenues for investigating topology reasoning in traffic scenes.

## 1   Introduction

In recent years, the availability of large-scale datasets and benchmarks has greatly facilitated research on autonomous driving. A critical aspect would be understanding the complex driving environment, which is the prerequisite for reasonable decisions. Many datasets [1, 10, 18, 44, 45] focus on perceiving visible lanelines to keep vehicles on the right track, while others [13, 14, 37, 39, 46] are specified in acquiring traffic information through detecting traffic signals. Nevertheless, this separation of tasks represents a limited understanding of the driving scene. For instance, when driving into a crossroad without any visible laneline, an autonomous vehicle might wonder which direction to go. Meanwhile, when a vehicle proceeds into an intersection where there is a green light presented, it is still possible that the traffic signal does not control the lane in which the car is driving. In this work, we build a strong association among traffic elements and lanes, aiming to create a topology relationship of the physical world and thus facilitate decision-making in the downstream tasks.

To keep autonomous vehicles driving in the correct position, the concept of lanes needs to be introduced. The perception of lanelines, which are the visible separation of lanes, is well explored. Previous datasets [1, 18, 25, 45] annotate lanelines on images in the perspective view. Such a 2D representation is insufficient to fulfill real-world requirements. When projecting 2D laneline into bird's-eye-view (BEV) space, lane direction would diverge/converge if the height dimension is ignored, leading to improper action decisions in the planning and control module in challenging

---

[*]Equal Contribution

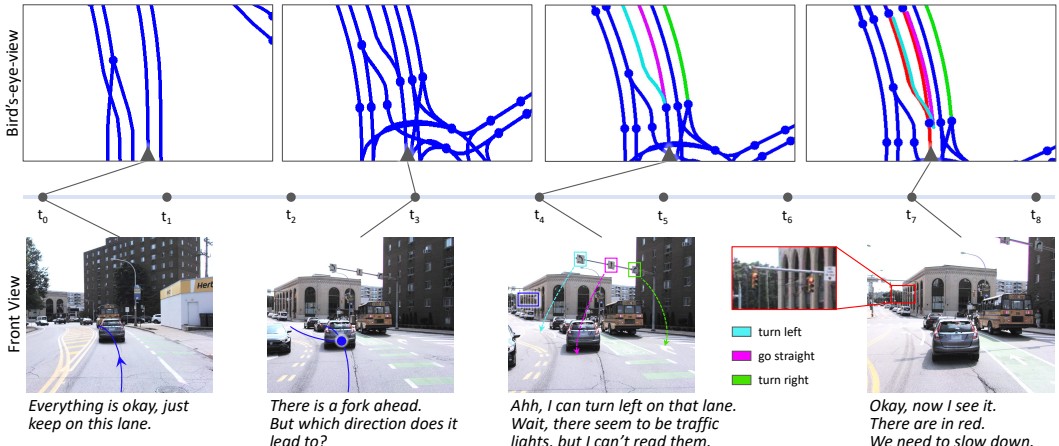

Figure 1: **Motivation and Overview of OpenLane-V2.** The dataset comprises various types of annotations, including instances and topology relationships. The directed centerlines provide trajectories for self-driving cars, and their connectivities build the lane network. Traffic elements with semantic labels deliver real-time traffic information. The associations between centerlines and traffic elements imply that a traffic element controls some particular lanes based on traffic rules.

scenarios. Recent works [8, 10, 44] define lanelines in the 3D space but still limit the labeling range within the front-view image. Studies on HD map learning [27, 28] incorporate multi-view images to perceive visible road entities, namely lanelines, road boundaries, and pedestrian crossings. However, serving as separations of neighboring lanes, the visible lanelines might not benefit downstream tasks directly. In common circumstances, vehicles follow the center of lanes, *i.e.*, lane centerlines, to drive on the road. To generate this type of invisible and conceptual trajectories, post-processing techniques are required based on the perceived lanelines. However, the desired trajectory becomes empty and the vehicle loses guidance when lanelines are absent, such as driving into a crossroad that typically does not have markings.

Similarly, the perception of traffic signals is formulated as a classic 2D detection problem on front-view images. Though traffic elements on the roads, such as traffic lights and road signs, provide practical and real-time information, existing formulations [13, 14] emphasize the accuracy of their positions but ignore proper guidance for cars on the road. The reason is that one traffic signal may control one or several lanes according to predefined traffic rules. Given that all traffic elements within a scene are perceived simultaneously, there exists the possibility for vehicles to be confused about which is the appropriate traffic instruction to obey. Hence, topology relationships between centerlines and traffic elements are established to assign traffic information to a particular lane.

As depicted in Figure 1, we seek to unify the aforementioned tasks and provide a comprehensive understanding of driving scenes, including the static entities such as lanes and traffic elements, along with their topology relationships. To this end, we propose the OpenLane-V2 dataset to shed light on the task of **scene structure perception and reasoning**. The requirement of perception is to obtain correct instance-level information, such as positions and semantic meanings, from captured scenes, while reasoning is to deduce topology relationships of perceived entities to generate a reasonable understanding of the environment. For the newly defined task, we strive to make our metric capable of covering all aspects of the task. The **OpenLane-V2 Score (OLS)** summarizes model performances with its component of DET and TOP scores for perception and reasoning respectively. Section 4 describes the proposed tasks and metrics.

Inherited from the OpenLane dataset [8], which is the first real-world and large-scale 3D lane dataset, **OpenLane-V2**, provides lane annotations in 3D space to reflect their properties in the real world. The directed lane centerlines and their connectivity serve as map-like perception results to facilitate downstream tasks. In addition to the annotations of traffic elements, we establish relationships between centerlines and traffic elements. That is, the correspondence between a lane and a traffic element is denoted as valid if and only if the traffic element controls the lane. With these representations, self-driving vehicles understand the current driving scenarios and know where to go or whether to accelerate. For more details on the proposed dataset, please refer to Section 3.

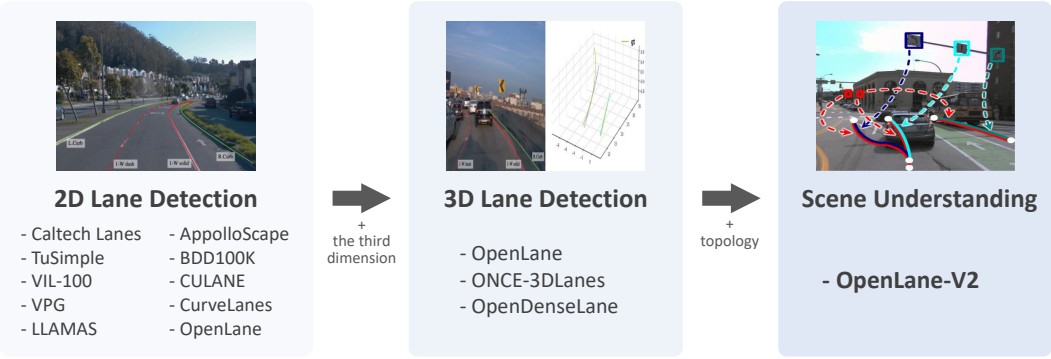

Figure 2: **Roadmap of lane detection datasets.** Most of the previous works provide only 2D labels. Benefiting from the pioneering OpenLane dataset [8], lane annotations in 3D space have gained great popularity in recent years. Taking one step further, the OpenLane-V2 dataset extends the annotation range of 3D lanes to encompass multi-view images and includes topology relationships to promote the task of scene understanding.

To sum up, our contributions are as follows:

- We present the OpenLane-V2 dataset for benchmarking the task of scene understanding. To the best of our knowledge, OpenLane-V2 is the first dataset that focuses on topology reasoning in the autonomous driving domain. Tasks and corresponding metrics are dedicatedly designed to evaluate model performance on the proposed benchmark.

- Built on top of awesome benchmarks, OpenLane-V2 includes massive images collected from various cities worldwide. It contains 2.1M instance-level annotations and 1.9M positive topology relationships. All annotations are carefully validated.

- We provide a development kit for easy access to the proposed dataset. Besides, plug-ins to prevail deep learning frameworks for training models would be jointly maintained with the community. The test server and leaderboard will also be maintained for fair comparisons.

## 2 Related Work

### 2.1 3D Lane Detection

The task of lane detection has been pursued for several years (Figure 2). Previous works [1, 2, 25, 40, 42, 43] provided 2D laneline annotations in the perspective view. CULANE [31] collected a large scale of data and manually annotated the occluded lane markings with cubic splines. With multiple sensors, AppolloScape [18] included per-pixel lane mark labeling in 35 classes. BDD100K [45] labeled lanes attributes of continuity (full or dashed) and direction (parallel or perpendicular) on a massive amount of data. However, the scope of annotation is still limited in the 2D space on front-view images. OpenLane [8] was the first large-scale, real-world 3D laneline dataset. It is equipped with a wide span of diversity in both data distribution and task applicability. In the spirit of it, the OpenLane-V2 dataset provides 3D annotations of lanes, which cover the whole surrounding area of the ego vehicle. Instead of focusing on the visible lanelines, annotations of conceptual centerlines in the proposed dataset serve as the trajectory guidance for downstream tasks. Moreover, as human drivers also observe situations from backward, we provide lane annotations in all directions of the ego car within a long range.

### 2.2 Traffic Element Recognition

Over the last decade, existing datasets have annotated traffic elements on images from the driving scenarios. Table 1 summarizes the relevant counterparts. Most of the works in the early 2010s [4, 16, 30, 37, 39] comprised a small amount of data. GTSRB [37] collected data from multiple German landscapes and showed that neural networks could outperform human test persons in detecting traffic signs. MTSD [13] made a step forward in both scale and diversity that contained 100K street-level

Table 1: **Comparison of current traffic element datasets.** "# Img.", "# Cls.", and "# Anno." denote the number of images, classes, and annotations respectively. "Track." implies that a traffic element has a unique tracking ID in different frames. "Corr." indicates whether the correspondences between lanes and traffic elements are annotated. * We decompose semantic labels of traffic elements into base attributes and omit elements that require OCR to acquire their meanings for vision-centric perception.

| Dataset | # Img. | # Cls. | # Anno. | Track. | Corr. | Resolution | Region | Year |
|---|---|---|---|---|---|---|---|---|
| LaRA [11] | 11K | 4 | 9K | ✓ | ✗ | 640×480 | France | 2009 |
| Stereopolis [4] | 847 | 4 | 251 | ✗ | ✗ | 960×1080 | France | 2010 |
| GTSRB [37] | 5K | 43 | 39K | ✓ | ✗ | 1360×1024 | Germany | 2012 |
| LISA [30] | 6K | 49 | 7K | ✗ | ✗ | 1280×960 | USA | 2012 |
| GTSDB [16] | 900 | 43 | 1K | ✗ | ✗ | 1360×800 | Germany | 2013 |
| BelgiumTS [39] | 9K | 62 | 13K | ✗ | ✗ | 1628×1236 | Belgium | 2013 |
| RTSD [34] | 179K | 156 | 104K | ✓ | ✗ | - | Russia | 2016 |
| TT100K [46] | 100K | 221 | 26K | ✗ | ✗ | 2048×2048 | China | 2016 |
| BSTLD [3] | 13K | 15 | 24K | ✓ | ✗ | 1280×720 | USA | 2017 |
| DTLD [14] | - | 344 | 230K | ✗ | ✗ | 2048×1024 | Germany | 2018 |
| MTSD [13] | 100K | 313 | 325K | ✗ | ✗ | - | Worldwide | 2020 |
| OpenLane-V2 | 466K | 13* | 258K* | ✓ | ✓ | - | Worldwide | 2023 |

images worldwide from diverse scenes, geographical locations, and varying weather and lighting conditions. Though with dedicated labels, previous datasets mainly pay attention to the correct location of traffic elements and are limited in the understanding of traffic elements. In addition to the positional label of traffic elements, we provide annotations on topology relationships of presented objects, enabling autonomous vehicles to have an understanding of the driving environment.

## 2.3 Scene Understanding

Understanding the driving scene plays a vital role in autonomous driving, especially in complicated scenarios. Few datasets focus on the comprehension of captured scenes. Current datasets [19, 23, 24, 29] comprised 2D images on which there are only a small amount of objects. Datasets in the human-object interaction domain [7, 15] limited the labeled relationship to the interactions between human beings and detected objects. The aforementioned datasets include annotations such as "cat-ride-snowboard", which are relationships between closely located objects. However, in our case, a traffic light may correspond to a lane in the distance rather than a closer one. To predict the correct relationships, models are required to have an understanding of the predefined traffic rules.

In the field of autonomous driving, previous works try to understand the intention of perceived entities. Tian *et al.* [38] provided pairwise relationships between moveable objects, *e.g.*, vehicles and pedestrians. Singh *et al.* [36] defined events as triplets, which comprise agents with their actions and locations. Other datasets paid attention to the intention and future behavior of driver [21, 32] or non-driver [22, 33] agents. While most of the existing tasks focus on the behavior of foreground movable objects, understanding the static background is also important for the downstream planning module [9, 17, 20, 35]. In this work, we emphasize the understanding of the driving scene, which provides trajectory information for self-driving vehicles.

## 3 OpenLane-V2

In this section, we give an overview of the OpenLane-V2 dataset, which is publicly available in our repository. Built on top of the Argoverse 2 [41] and nuScenes [5] datasets, which are both distributed under the CC BY-NC-SA 4.0 license, the proposed dataset includes images in 2,000 scene segments collected worldwide under different challenging environments, covering noon and night, sunny and rainy days, downtown and suburbs. Based on the provided HD maps and through a dedicated labeling process, we deliver high-quality annotations with the help of experienced annotators and multiple validation stages. The proposed dataset is under the CC BY-NC-SA 4.0 license, while the code is under the Apache License 2.0.

Table 2: **Statistics of OpenLane-V2.** All frames are accompanied by annotations. The annotation range is larger to the front and back compared to that in current methods, which is commonly set to $\pm 30m$. # is an abbreviation for the number of. * Front-view images are transposed to $1550 \times 2048$.

| | $subset\_A$ | | | $subset\_B$ | | |
|---|---|---|---|---|---|---|
| Split | Train | Val | Test | Train | Val | Test |
| Sample Rate | | | | $2Hz$ | | |
| Annotation Range | | | $\pm 50m$ (x-axis), $\pm 25m$ (y-axis) | | | |
| # Camera | | 7 | | | 6 | |
| Image Resolution | | $2048 \times 1550*$ | | | $1600 \times 900$ | |
| Avg. Duration of Scene Segments | | $15s$ | | | $20s$ | |
| # Scene Segment | 700 | 150 | 150 | 700 | 150 | 150 |
| Avg. # Centerline per Frame | 26.34 | 26.44 | 26.50 | 24.32 | 24.80 | 23.82 |
| Avg. # Traffic Element per Frame | 3.70 | 3.69 | 2.80 | 3.58 | 3.76 | 3.25 |
| Avg. # Connection per Centerline | 1.90 | 1.89 | 1.89 | 1.83 | 1.79 | 1.84 |
| Avg. # Corresponded Centerline per Traffic Element | 0.71 | 0.83 | 0.91 | 0.54 | 0.52 | 0.58 |

## 3.1 Raw Data Acquisition

As camera-centric methods attract a large amount of attention in academia and industry, we incorporate multi-view images from original datasets. Due to differences in sensor setups, whereby image data is independently collected in Argoverse 2 [41] and nuScenes [5], we divide the proposed dataset into $subset\_A$ and $subset\_B$ respectively, as described in Table 2. The $subset\_A$ comprises scenes from six cities: Austin (3.1%), Detroit (11.7%), Miami (35.4%), Pittsburgh (35.0%), Palo Alto (2.2%), and Washington D.C. (12.6%), while the $subset\_B$ is collected from two cities: Boston (55.0%) and Singapore (45.0%). The $subset\_A$ includes 3.0% night scenes and 1.1% rain scenes, while the $subset\_B$ includes 11.7% night scenes and 17.4% rain scenes. Despite discrepancies in camera settings, the coordinate system is unified and right-handed. For ego coordinate, the x-axis is positive forwards, the y-axis is positive to the left, and the z-axis is positive upwards. Camera intrinsics, extrinsics, and ego-vehicle poses in the global coordinate system are provided.

## 3.2 Centerlines and Their Connectivity

In the provided HD maps, map elements are represented as lane segments, containing boundary, mark type, neighbors, predecessors, successors, *etc*. The problem is that lanelines are divided based on rules for constructing HD maps but not visually apparent marks, as the primary objective is to map the world rather than facilitate autonomous driving directly. This characteristic introduces unnecessary noise and hinders the learning process of models. In this work, we represent a single lane as an instance. To generate the ground truth of centerlines, we first regress their locations using the boundary information from HD maps. We then merge lanes with only one predecessor or successor to ensure the continuity of lanes. Lanes are separated into different instances if and only if in the cases of intersection, fork, and merge. Topology relationship is then provided on the merged lanes.

The annotation of a centerline is provided in 3D space through an ordered list of points. Specifically, for a centerline $[p_1, ..., p_n]$, $p_1 = (x_1, y_1, z_1)$ represents the starting point of the lane, while $p_n = (x_n, y_n, z_n)$ denotes the ending point. Note that the ego car is located at $(0, 0, 0)$ for each frame, and values of the z-axis of $subset\_B$ are set to 0, as its HD maps exclude the height information. We set $n$ to 201 in the given data, but subsample 11 points for each lane for efficient evaluation. The direction of a centerline, from the starting point to the ending point, denotes that a vehicle should follow the direction when driving on this lane based on the predefined traffic rules. Topology relationships are provided as adjacency matrixes for each frame based on the ordering of centerlines. Since a lane is directed and represented as a list of points, the connection of two lanes means that the ending point of a lane is connected to the starting point of another lane. Statistically, about 90% of frames have more than 10 centerlines, while about 10% have more than 40. Most lanes have one predecessor or successor, but in complex scenarios such as crossroads, the number can be up to 7.

## 3.3 Traffic Elements and Their Correspondence to Centerlines

Traffic elements, such as traffic lights, road markings, and road signs, provide valuable instructions for autonomous vehicles. As critical traffic elements are usually exhibited in the front view, and

their accurate 3D locations are not required for guiding autonomous vehicles, we only annotate them in 2D format on the front-view images. Each traffic element is annotated with a 2D bounding box $(x_1, y_1, x_2, y_2)$, where $(x_1, y_1)$ is the top-left corner and $(x_2, y_2)$ is the bottom-right corner. Additionally, we label the attributes of each traffic element. In detail, the attribute of elements, whose semantic meaning is unobservable, is set to *unknown*, while valid elements are annotated as *red*, *green*, *yellow*, *go_straight*, *turn_left*, *turn_right*, *no_left_turn*, *no_right_turn*, *u_turn*, *no_u_turn*, *slight_left*, or *slight_right*, resulting in 13 various attributes in total. Note that those traffic elements having composite attributes would be divided into multiple annotations with decomposed attributes sharing the same bounding box. For instance, a road sign, which is at the position of $(x_1, y_1, x_2, y_2)$ and with the meaning of "go straight and turn left", is divided into two bounding boxes, namely $(x_1, y_1, x_2, y_2, go\_straight)$ and $(x_1, y_1, x_2, y_2, turn\_left)$. The class imbalance should be noticed in that specific traffic elements, such as *u_turn*, are much rarer than the common ones like traffic lights in red or green.

The correspondence of a lane and a traffic element forms a regulation for vehicles driving in a particular lane. The construction of relationships between spatially relevant lanes and traffic elements is straightforward. For instance, a lane is controlled by the road marking which is located within its boundary. However, most of the centerlines and traffic elements do not fit into this case. We utilize the following principles for the labeling process. For those traffic element which does not contain directional information, it is associated with centerlines on which only going straight is permitted. Traffic elements with directional information, such as traffic lights in the shape of a left arrow, control the corresponding lanes going in the same direction. Note that traffic elements are only associated with centerlines outside the intersection.

## 4 Task Definition & Evaluation Metric

In this section, we introduce the tasks and metrics in OpenLane-V2. The primary task of the proposed benchmark is scene structure perception and reasoning, which requires the model to recognize lanes and their dynamic drivable states in the surrounding environment. The challenge includes detecting lane centerlines and traffic elements, recognizing the attributes of traffic elements, and reasoning about the topology relationships on perceived entities. We further divide the primary task into three subtasks: 3D lane detection, traffic element recognition, and topology recognition. The OpenLane-V2 Score (OLS), which is the average of various metrics from different subtasks, is defined to describe the overall performance of the primary task:

$$\text{OLS} = \frac{1}{4}\left[\text{DET}_l + \text{DET}_t + f(\text{TOP}_{ll}) + f(\text{TOP}_{lt})\right], \tag{1}$$

where $f$ is a scale function to emphasize the task of topology reasoning.

### 4.1 3D Lane Detection

In the spirit of the OpenLane dataset [8], which is the first real-world and the largest scaled 3D lane dataset to date, we provide lane annotations in 3D space. We define the subtask of 3D lane detection as perceiving directed 3D lane centerlines from the given multi-view images covering a fully panoramic field-of-view (FOV).

Given a pair of curves, namely a ground truth $v_l = [p_1, ..., p_n]$ and a prediction $\hat{v}_l = [\hat{p}_1, ..., \hat{p}_k]$, their geometric similarity is measured by the discrete Fréchet distance [12]. Specifically, a coupling $L$ is defined as a sequence of pairs between points in $v$ and $\hat{v}$:

$$(p_{a_1}, \hat{p}_{b_1}), ..., (p_{a_m}, \hat{p}_{b_m}), \tag{2}$$

where $1 = a_1 \le a_i \le a_j \le a_m = n$ and $1 = b_1 \le b_i \le b_j \le b_m = k$ for all $i < j$. Then the norm $||L||$ of a coupling $L$ is defined as the distance of the most dissimilar pair in $L$. The Fréchet distance of a pair of curves is the minimum norm of all possible coupling:

$$D_{\text{Fréchet}}(v_l, \hat{v}_l) = min\{||L|| \mid \forall \, possible \, L\}. \tag{3}$$

We define a threshold $t \in \mathbb{T}$ that a pair of centerlines would be regarded as unmatched if their distance is greater than $t$. Then $\text{DET}_l$ is averaged over match thresholds of $\mathbb{T} = \{1.0, 2.0, 3.0\}$:

$$\text{DET}_l = \frac{1}{|\mathbb{T}|} \sum_{t \in \mathbb{T}} AP_t. \tag{4}$$

The $AP$ score is the area under the precision-recall curve, defined as $\int_0^1 p(r)\mathrm{d}r$, where $p$ and $r$ denote precision and recall respectively. Note that as the defined annotation range is relatively large compared to previous datasets, accurate perception of lanes in the distance would be challenging. Thus, the matching thresholds are relaxed for centerlines at a distance based on the distance between the ground truth lane and the ego car. For instance, a lane at a distance would be thresholded on $1.2t$ while another lane in a closer region would be thresholded on $1.1t$ for the same threshold $t \in \mathbb{T}$.

### 4.2 Traffic Element Recognition

Traffic elements and their attribute provide crucial information for autonomous vehicles. The attribute represents the semantic meaning of a traffic element, such as the red color of a traffic light. In this subtask, on the given image in the front view, the location of traffic elements and their attributes are demanded to be perceived simultaneously. Compared to typical 2D detection datasets, the challenge is that the size of traffic elements is tiny due to the large scale of outdoor environments.

To preserve consistency to the distance mentioned above, we modify the common IoU (Intersection over Union) measure as a distance that:

$$D_{IoU}(v_t, \hat{v}_t) = 1 - \frac{|v_t \cap \hat{v}_t|}{|v_t \cup \hat{v}_t|}, \tag{5}$$

where $v_t$ and $\hat{v}_t$ are the ground truth and predicted bounding box respectively. We consider IoU distance as the affinity measure with a match threshold of $0.75$. The $\mathrm{DET}_t$ score is utilized to measure the performance of traffic elements detection and is averaged over different attributes $\mathbb{A}$ that:

$$\mathrm{DET}_t = \frac{1}{|\mathbb{A}|} \sum_{a \in \mathbb{A}} AP_a. \tag{6}$$

### 4.3 Topology Reasoning

We first define the task of recognizing topology relationships in the field of autonomous driving. On the perceived entities, the topology relationships are built. For simplicity, we divide the graph on all entities into two subgraphs. The connectivity of directed lanes establishes a map-like network and is denoted as the lane graph $(V_l, E_{ll})$. Note that the edge set $E_{ll} \subseteq V_l \times V_l$ is asymmetric, as the incoming and outgoing edges of a lane represent the connection on its starting and ending points respectively. An entry $(i, j)$ in $E_{ll}$ is positive if and only if the ending point of the lane $v_i$ is connected to the starting point of $v_j$. Besides, the undirected graph $(V_l \cup V_t, E_{lt})$ describes the correspondence between centerlines and traffic elements. It can be seen as a bipartite graph that positive edges only exist between $V_l$ and $V_t$.

The TOP score, which is an mAP metric adapted from link prediction in the graph domain, is utilized to evaluate model performance on the matched graphs. Given a ground truth graph $G = (V, E)$ and a predicted one $\hat{G} = (\hat{V}, \hat{E})$, it is possible that the number of predicted vertices is not equal to that of ground truth. We first build a projection between predictions and ground truth to preserve true positive vertices, according to the entity-specific similarity measures, namely Fréchet and IoU distances for centerlines and traffic elements respectively. The resulting vertex set $\hat{V}'$ needs to fulfill the requirements such that $V = \hat{V}'$ and $\hat{V}' \subseteq \hat{V} \cup \{v_d\}$, where $\{v_d\}$ is a set of dummy vertices. Then the TOP score is the averaged mAP between $(V, E)$ and $(\hat{V}', \hat{E}')$ over all vertices:

$$\mathrm{TOP} = \frac{1}{|V|} \sum_{v \in V} \frac{\sum_{\hat{n}' \in \hat{N}'(v)} P(\hat{n}') \mathbb{1}(\hat{n}' \in N(v))}{|N(v)|}, \tag{7}$$

where $N(v)$ denotes the neighbors of vertex $v$, $P(v)$ is the precision of vertex $v$ in the ordered list ranked by predicted confidences, and positive edges are those whose confidence is greater than $0.5$. The $\mathrm{TOP}_{ll}$ is for topology among centerlines on the graph $(V_l, E_{ll})$, while the $\mathrm{TOP}_{lt}$ is for topology between lane centerlines and traffic elements on the graph $(V_l \cup V_t, E_{lt})$.

Table 3: **Quantitative results** on the OpenLane-V2 *val* split. It is observed that the model design of how to represent centerlines in the network has an impact on model performance. "Instance" denotes that a centerline is represented as a single query in the network, while "Point Set" indicates that a centerline is described by a set of independent points.

| Method | Design | subset_A | | | | | subset_B | | | | |
| | | OLS | $DET_l$ | $DET_t$ | $TOP_{ll}$ | $TOP_{lt}$ | OLS | $DET_l$ | $DET_t$ | $TOP_{ll}$ | $TOP_{lt}$ |
|---|---|---|---|---|---|---|---|---|---|---|---|
| STSU [6] | Instance | 25.4 | 12.7 | 43.0 | 0.5 | 15.1 | 21.2 | 8.2 | 43.9 | 0.0 | 9.4 |
| VectorMapNet [28] | Point Set | 20.8 | 11.1 | 41.7 | 0.4 | 5.9 | 16.3 | 3.5 | 49.1 | 0.0 | 1.4 |
| MapTR [27] | Point Set | 20.0 | 8.3 | 43.5 | 0.2 | 5.9 | 21.1 | 8.3 | 54.0 | 0.1 | 3.7 |
| TopoNet [26] | Instance | 35.4 | 29.2 | 48.0 | 4.1 | 19.3 | 33.2 | 24.3 | 55.0 | 2.5 | 14.2 |

## 5 Experiments

In this section, we adapt and evaluate multiple state-of-the-art methods on the proposed OpenLane-V2 dataset. Visualizations and analysis are then reported to investigate the impact of different design choices on model performance.

### 5.1 Baselines

In our experiments, various models are involved, including STSU [6], VectorMapNet [28], MapTR [27], and TopoNet [26]. STSU utilizes a DETR-like neural network to detect centerlines and then derive their connectivity by a successive MLP module. Since it is designed for monocular image inputs and contains the BEV representation as intermedia results, we adapt it to multi-view inputs by concatenating BEV embeddings from different views. VectorMapNet directly represents each map element as a sequence of points and predicts positional information in an auto-regressive manner. MapTR models each map element as a point set with a group of equivalent permutations to deal with geometrical ambiguity. For both methods, the BEV range is expanded to fit dataset requirements. As for training, we supervise VectorMapNet and MapTR with centerlines and use the element queries in the Transformer decoder as instance queries to produce topology relationships. TopoNet is specifically designed for the task of scene structure understanding that it utilizes instance-level queries for both centerlines and traffic elements to handle their locations and topology relationships. Note that except for TopoNet, methods mentioned above are further appended with heads for predicting traffic elements and topology.

### 5.2 Results

Quantitative results on the OpenLane-V2 dataset are illustrated in Table 3. It is not surprising that the $DET_l$ scores are low for all methods, since the task of centerline perception is challenging for existing networks. Different from detecting visible lanelines, the centerline is invisible, which requires models to deduce their position by obtaining references from the neighboring lanelines. Additionally, the direction of centerlines is determined by the entire scene, adding extra complexity to the task. For traffic elements recognition, the tiny size of traffic elements, as depicted in Figure 3, also introduces difficulty for the models. As the TOP scores require a match between ground truths and predictions, performance on the connectivity between centerlines is not ideal due to the unsatisfactory perception results of centerlines. To increase model performance on topology reasoning, it is a must to generate perception results that are sufficiently accurate compared to the ground truth.

In Figure 3, we present visualizations of predicted results in a complex crossroad. Although all models provide an approximate shape of the intersection, their performance on the position and shape of centerlines, as well as their topology relationship, can still be improved. STSU generates a large number of false positive centerline predictions and fails to attach traffic information to centerlines. As MapTR represents points of a lane independently in the networks, the predicted centerlines are not in the shape of driving trajectories, which are commonly smooth. Meanwhile, although VectorMapNet utilizes a point set to describe a centerline as well, its auto-regressive design ensures a proper shape of centerlines. VectorMapNet, MapTR, and TopoNet can only provide partially correct results on the semantic information of lanes, namely the correspondence between centerlines and traffic elements. A potential solution could be introducing more prior, such as knowledge of traffic rules, to the network for reasoning about the association between traffic elements of a lane.

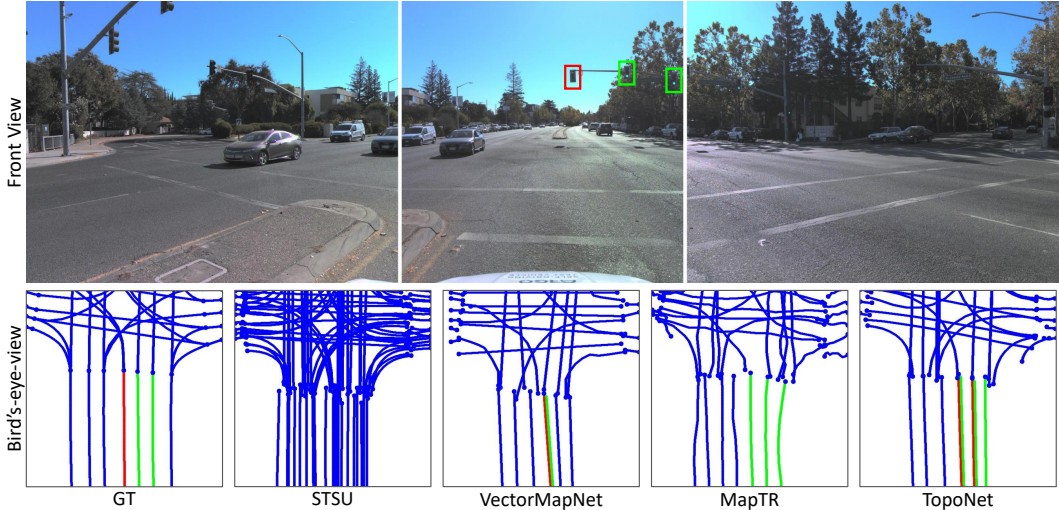

Figure 3: **Qualitative results** of various algorithms. Traffic lights in red and green are emphasized with red and green boxes respectively. The bird's-eye-view is truncated at $\pm25m$ along the x-axis.

## 6    Conclusion

In this paper, we introduce OpenLane-V2, aiming to facilitate the task of scene structure perception and reasoning. The lane network is represented by lane centerlines and their connectivity, while traffic information is described by traffic elements with semantic meanings and their association with lanes. Tasks and metrics are described in detail for future research usage. We adapt various methods and demonstrate their performance on the proposed dataset. We hope this dataset will encourage the research community to design and develop neural networks on the defined tasks, and further promote research in the field of autonomous driving.

**Limitations.** Due to limitations in available resources, the proposed dataset is built on top of existing datasets, and its data scale is the same as previous datasets. We believe that including more driving scenes will further increase its diversity. Moreover, although the lane networks with traffic information benefit the downstream tasks, we do not include the planning task in the proposed dataset, as it also requires knowledge of moveable objects, such as cars and pedestrians, for collision avoidance. We leave the integration of static and dynamic entities to future work.

**Impact.** Based on previous sections, it is evident that the released dataset is used for research purposes. Models trained or evaluated on this dataset should not be directly used for direct deployment or any real-world application. It should be noted that the proposed dataset does not provide any guarantee, particularly in safety-critical situations.

## Acknowledgements

This work was supported by National Key R&D Program of China (2022ZD0160104) and NSFC (62206172). We thank reviewers for their fruitful comments and the research community for participating in the OpenLane Topology Challenge 2023.

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
