# OpenLane-V2: Supplementary Material

## A   Overview

Our supplementary includes **author statement**, **licensing**, and **implementation details** of benchmark results for reproducibility. We also provide **dataset documentation** following the Datasheet format [3], including distribution, maintenance plan, composition, collection, and other details of our dataset.

## B   Author Statement

We bear all responsibilities for licensing, distributing, and maintaining our dataset.

## C   Licensing

The proposed dataset is under the CC BY-NC-SA 4.0 license, while the code in the repository is under the Apache License 2.0.

## D   Datasheet

### D.1   Motivation

**For what purpose was the dataset created?**   We propose this dataset in order to accurately depict the complex traffic scene and provide information for autonomous vehicles to execute accurate judgments. The dataset comprises various types of annotations, including instances and topology relationships. The directed centerlines offer trajectory guidance for self-driving cars, and their connectivities build the lane network. Traffic elements with semantic labels deliver real-time traffic information. The associations between centerlines and traffic elements imply that a traffic element controls some particular lanes based on traffic rules.

**Who created the dataset (e.g., which team, research group) and on behalf of which entity (e.g., company, institution, organization)?**   This dataset was created by Huijie Wang (OpenDriveLab), Tianyu Li (OpenDriveLab), Yang Li (OpenDriveLab), Li Chen (OpenDriveLab), Chonghao Sima (OpenDriveLab), Zhenbo Liu (Huawei), Bangjun Wang (OpenDriveLab), Peijin Jia (OpenDriveLab), Yuting Wang (OpenDriveLab), Shengyin Jiang (OpenDriveLab), Feng Wen (Huawei), Hang Xu (Huawei), Ping Luo (OpenDriveLab), Junchi Yan (OpenDriveLab), Wei Zhang (Huawei), and Hongyang Li (OpenDriveLab).

**Who funded the creation of the dataset?**   Shanghai AI Laboratory and Huawei.

### D.2   Distribution

**Will the dataset be distributed to third parties outside of the entity (e.g., company, institution, organization) on behalf of which the dataset was created?**   Yes, the dataset can be accessed publicly on the Internet.

**How will the dataset be distributed (e.g., tarball on website, API, GitHub)?**   The dataset is available at https://github.com/OpenDriveLab/OpenLane-V2.

### D.3 Maintenance

**Who will be supporting/hosting/maintaining the dataset?**    The authors will be supporting, hosting, and maintaining the dataset.

**How can the owner/curator/manager of the dataset be contacted (e.g., email address)?**    Please contact Huijie Wang (wanghuijie@pjlab.org.cn) and Hongyang Li (lihongyang@pjlab.org.cn).

**Is there an erratum?**    No. We will make a statement if there is any.

**Will the dataset be updated (e.g., to correct labeling errors, add new instances, delete instances)?**
Yes. New updates will be posted at `https://github.com/OpenDriveLab/OpenLane-V2`.

**Will older versions of the dataset continue to be supported/hosted/maintained?**    Yes.

**If others want to extend/augment/build on/contribute to the dataset, is there a mechanism for them to do so?**    Yes. Please refer to `https://github.com/OpenDriveLab/OpenLane-V2`.

### D.4 Composition

**What do the instances that comprise the dataset represent?**    An instance of the dataset represents a driving scene at a particular time point (time frame). Multi-view images provide a comprehensive view of the driving scene. Annotations for each frame are provided as follows: (1) Centerlines represent the midpoints of the lanes that vehicles should follow when driving on the road. (2) Traffic elements describe objects that contain information for drivers, such as traffic lights that indicate when to go or stop. (3) The topology relationship between centerlines represents the connectivity of the lanes. (4) The topology relationship between centerlines and traffic elements indicates that a vehicle driving in a particular lane must follow the traffic signal if and only if the topology between them exists.

**How many instances are there in total (of each type, if appropriate)?**    OpenLane-V2 consists of 2,000 annotated road scenes with a total number of 72K frames. Each scene contains a varying number of consecutive frames. Each scene includes multiple video frames that are annotated at a frequency of 2Hz. The dataset has approximately 1.8 million annotations of lane centerlines and about 0.25 million annotations of traffic elements.

**Are relationships between individual instances made explicit?**    Frames in a single segment are consecutively sampled at a rate of 2Hz. We offer annotations detailing the connectivity between lanes and the relationships between lanes and traffic elements. These annotations are provided as adjacency matrices for each frame.

**Are there recommended data splits (e.g., training, development/validation, testing)?**    We have already partitioned our dataset into three distinct splits: training, validation, and testing.

**Is the dataset self-contained, or does it link to or otherwise rely on external resources?**    This dataset is built on top of Argoverse 2 [8] and nuScenes [1], with additional annotations for the tasks defined by us. However, we have reconstructed and redistributed the raw data to ensure that the dataset is self-contained.

### D.5 Collection Process

**Who was involved in the data collection process (e.g., students, crowdworkers, contractors) and how were they compensated (e.g., how much were crowdworkers paid)?**    Based on the provided HD maps and through a labeling process, we deliver high-quality annotations with the help of experienced annotators and multiple validation stages. Annotations on traffic elements and topology relationships are made and verified by contractors.

### D.6 Use

**What (other) tasks could the dataset be used for?**   It comprises three primary sub-tasks, including 3D lane detection, traffic element recognition, and topology reasoning.

## E   Evaluation Metric

We provide an example of calculating the topological metrics. Given a graph with eight vertices, and vertex 1 whose ground-truth connected neighbors are {2, 3, 5, 8} and the predicted results are {4, 3, 5, 6, 7} sorted by confidences, we calculate the mAP score of the vertex 1 as follows:

$$
\begin{aligned}
mAP(1) &= \frac{FP(4) + TP(3) + TP(5) + FP(6) + FP(7)}{|\{2,3,5,8\}|} \\
&= \frac{0 \times 0 + 1 \times \frac{1}{2} + 1 \times \frac{2}{3} + 0 \times \frac{2}{4} + 0 \times \frac{2}{5}}{4} \\
&= \frac{7}{24},
\end{aligned}
\tag{1}
$$

where the first multiplier indicates if the predicted connection is correct or not, and the second one is the precision of the predicted neighbors. Supposed that the mAPs of other vertices are 0, the TOP score is averaged over all vertices:

$$
\begin{aligned}
TOP &= \frac{1}{8} \sum_{i=1}^{8} mAP(i) \\
&= \frac{\frac{7}{24} + 0 + 0 + 0 + 0 + 0 + 0 + 0}{8} \\
&= \frac{7}{192} \approx 0.036.
\end{aligned}
\tag{2}
$$

## F   Implementation Details

We adapt three state-of-the-art methods from centerline detection (STSU [2]) and map element detection (VectorMapNet [6] and MapTR [5]) into the OpenLane-V2 benchmark. TopoNet [4], which is specifically designed for the task of scene structure understanding, is also involved.

We train all models on both $subset\_A$ and $subset\_B$ of OpenLane-V2, and maintain consistency in the input settings across all methods. For $subset\_A$, the resolution of input images is $2048 \times 1550$, except for the front-view image, whose resolution is $1550 \times 2048$. The front-view image is cropped to $1550 \times 1550$, and zero padding is adopted to align the input resolution. For $subset\_B$, the resolution of input images is $1600 \times 900$. For data augmentation, resizing with a factor of 0.5 and color jitter is used by default. Training is performed on a cluster with 8 NVIDIA A100 GPUs. Typically, a 24-epoch training run consumes approximately 16 hours.

**TopoNet.** We use ResNet-50 as the backbone, and the output feature is from the stages of C3, C4, and C5. The size of BEV grids is set to $200 \times 100$. The loss weight of $\lambda_{cls_{TE}}$, $\lambda_{reg_{TE}}$, $\lambda_{iou_{TE}}$, $\lambda_{cls_{LC}}$, and $\lambda_{reg_{LC}}$ is set to 1.0, 2.5, 1.0, 1.5, and 0.025 respectively.

**STSU** employs EfficientNet-B0 [7] as the backbone. The model leverages a BEV positional embedding and a DETR head to predict centerlines, and uses object queries in the decoder to predict the connectivity of centerlines. We replace the original backbone with ResNet-50. To adapt to the multi-view inputs, we reimplement STSU, where we compute and concatenate the BEV embedding of images from different views. The concatenated embedding is then fed into the DETR encoder for further processing. We preserve the original DETR decoder to predict the Bezier control points, which are subsequently interpolated into 11 equidistant points. For the relationship between centerlines, the prediction head of STSU is preserved. For the traffic element detection branch and head for topology between lane and traffic elements, we employ the identical design as in TopoNet.

**VectorMapNet** uses a DETR-like decoder to estimate key points, complemented by an auto-regressive module to generate detailed positional information for a centerline instance. In our implementation, the backbone setting is aligned with TopoNet. We expand the perception range to $\pm50m \times \pm25m$ instead of $\pm30m \times \pm15m$. We also interpolate the centerline outputs of VectorMapNet to fit our setting during the prediction process. For topology prediction, we use the key point object queries in the decoder as instance queries of centerlines. We retain most of the settings of the open-source codebase of VectorMapNet. Due to its lack of support for 3D centerlines, we only predict 2D centerlines within the BEV space and ignore the height dimension during evaluation.

**MapTR.** We align the backbone setting with TopoNet. The area of perception is increased to $\pm50m \times \pm25m$. We set the number of point queries per lane to 11. In terms of topology prediction, the average of point queries of an instance in the MapTR decoder is used as the object query of a centerline. The traffic element head and topology head are configured in the same setting as in TopoNet. The implementation is based on the open-source codebase of MapTR, and we predict only 2D centerlines in BEV and disregard the z dimension during inference, following the same approach as in the VectorMapNet reimplementation.

## G   Illustrations

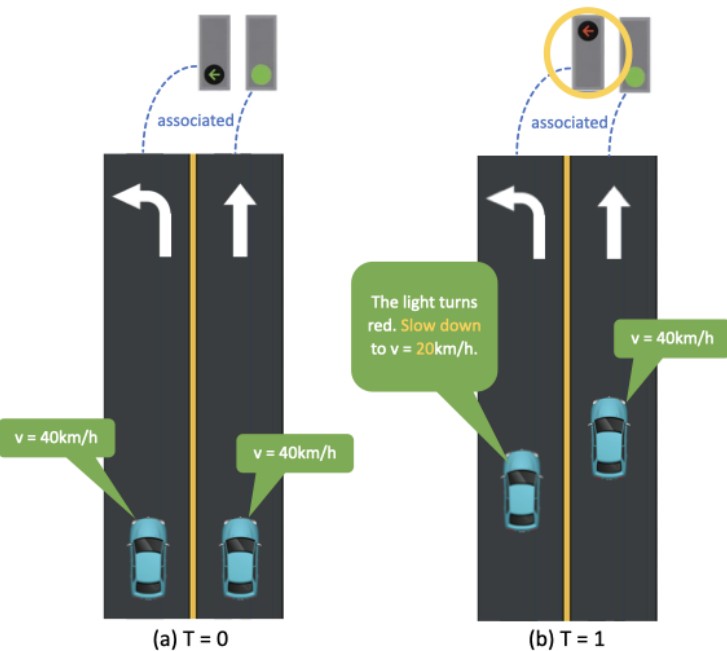

Figure 1: Illustration of how associations between centerlines and traffic elements benefit downstream tasks. In this example, the left vehicle on the left-turn lane needs to slow down as the traffic light turns red when $T = 1$, while the other vehicle could keep going straight at the same speed. Vehicles make their decisions according to the attribute assigned to the specific centerline. We focus on building the topology relationships to benefit inferring the corresponding attribute in this paper.

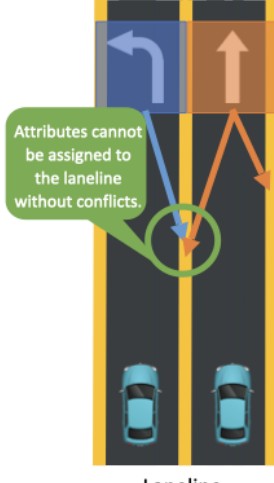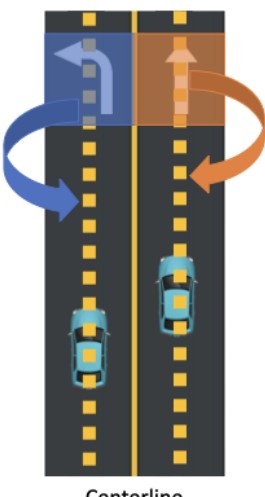

Laneline                    Centerline

Figure 2: Comparison of attribute assignment between lanelines and centerlines. Properties can be attached to centerlines, such as the lane direction. In traditional frameworks, the assignment is realized with two detection models (laneline detection and road elements detection) and complex hand-crafted rules.

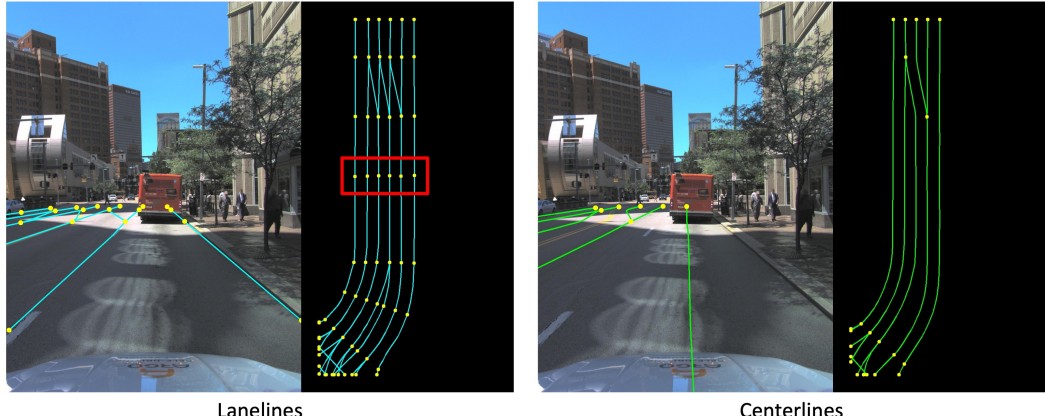

Lanelines                    Centerlines

Figure 3: Comparison of the ambiguity between lanelines and centerlines. Centerlines are only divided in some particular positions with apparent indications, such as splitting and merging of lanes, while lanelines are divided in some visually continuous positions (*i.e.*, the red box in the left figure).

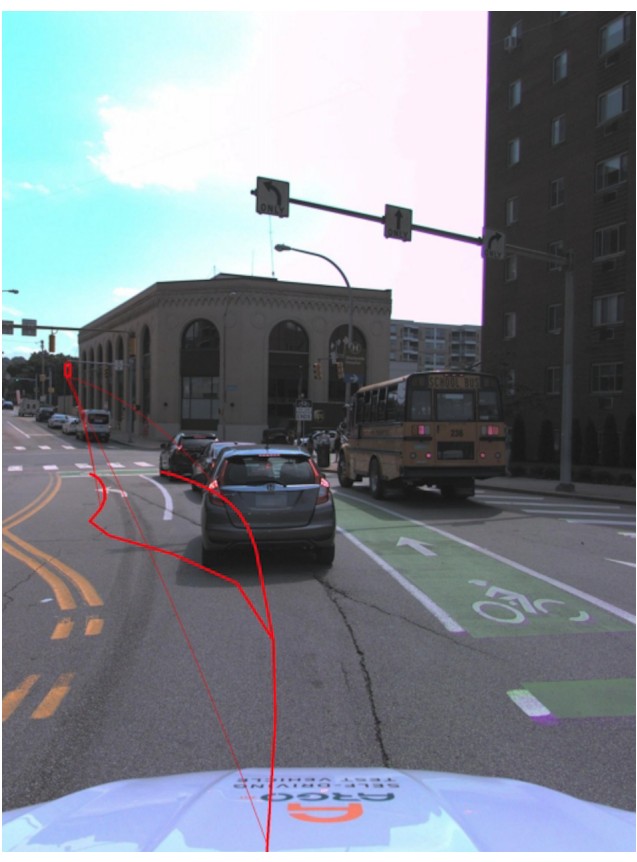

Figure 4: Example of "many to many" mapping in the dataset. The traffic element in the red box is associated with three centerlines.

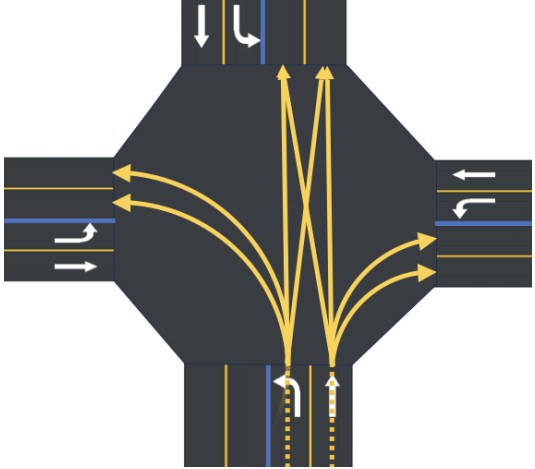

Figure 5: Illustration of centerlines in an intersection. The complex nature of multi-directional connections in an intersection leads to a high number of centerlines.