# OpenReview forum: "OpenLane-V2: A Topology Reasoning Benchmark for Unified 3D HD Mapping"
_NeurIPS.cc/2023/Track/Datasets_and_Benchmarks — NeurIPS 2023 Datasets and Benchmarks Poster_

### Official Review · Reviewer_bqmb · 2023-07-20
**Good dataset for scene structure perception and reasoning**

**Rating:** 8
**Confidence:** 4

**Strengths:**

- Creating a dataset that couples visible lane perception and traffic signals enable a richer scene understanding than treating them separately. Understanding the topologic relations of the roads and traffic elements in the physical world allows the development of models that simplify decision-making in the downstream tasks.
- OpenLane-V2 made use of well-known, publicly available datasets. This solution enables the easy utilization of the proposed dataset for the research community.
- The association of the traffic elements with corresponding centerlines (i.e., scene structure perception and reasoning) is a vital problem of autonomous driving without publicly available datasets built for this purpose. This problem might be alleviated with OpenLane-V2.
- The centerline is a good representation.
- Real-world, large-scale dataset (2000 scenes, 2.1M instance-level annotations, 1.9M topology relationships).


**Additional Feedback:**

Please provide some description of the validation stages of manual annotation.

**Clarity:**

The paper is well-written and easy to follow, with sufficient figures and tables.

**Correctness:**

The claims made by the authors are sound. The dataset construction is partly done by the authors of nuScenes and Argoverse 2. The manual annotation process and especially the quality check of these annotations shall be described more in-depth though.
The task definitions and proposed metrics are described in detail. The baseline models (with further implementation details in the supplementary material) and experimental results are provided by the authors.

**Documentation:**

The documentation contains sufficient information required for acceptance. Furthermore, the repository has a detailed description of the dataset and defined tasks. The devkit provides the community easy access to the dataset with an example Jupyter notebook for visualizing the data. The baseline trainings are reproducible based on the description on the repository. The large number of stars on the GitHub repository and the participants in the contest indicate that the research community already utilizes the proposed dataset.

**Ethics:**

I did not find any ethical concerns. Since the dataset is built on publicly available datasets, consent, and privacy issues are not problematic. Furthermore, responsible use and legal compliance are mentioned in the Impact part of Section 6.

**Limitations:**

The authors indicated the limitations of their work (class imbalance in traffic elements, diversity needs to be increased, planning task is not included in the dataset) as well as the possible negative societal impacts of their work. The comments in the 'Opportunities for Improvement' part of the review that are not covered by the 'Limitations' section in the paper might result in improvements in the dataset.

**Opportunities For Improvement:**

- A detailed description of the multiple validation stages of manual annotation would help to understand how high-quality annotation is ensured.
- The annotation range (+/-50 meters along the x-axis) might be too short for real-time applications.
- The data is recorded mainly in the US. It's not clear whether models trained on OpenLane-V2 will be able to generalize to different countries.
- Traffic elements are annotated in 2D only. The missing 3D information might help in the centerline-traffic element association for the models.

**Relation To Prior Work:**

The authors provided a comprehensive literature review, clearly describing the similarities and differences between the prior art and the proposed work. In addition, relevant citations are included.

**Summary And Contributions:**

The submission introduces the first dataset on topology reasoning for traffic scene structure. The authors built on existing datasets (nuScenes, Argoverse2) and extended them to describe traffic elements and their correlations to the lanes utilizing the expertise of manual annotators. Furthermore, two tasks have been introduced besides the already available conventional 3D lane perception task: traffic element recognition and topology reasoning in the autonomous driving domain. In addition, the OpenLane-V2 Score is introduced to summarize model performances for perception and reasoning. Finally, several state-of-the-art models have been evaluated quantitatively and qualitatively on the proposed OpenLane-V2 dataset.

---

> ### Author Response · Authors · 2023-08-12
> **Author response to Reviewer bqmb**
>
> Dear Reviewer bqmb,
>
> Thank you for commenting on the strengths and contributions of our dataset. We address your questions below.
>
> > ***Q1: A detailed description of the multiple validation stages of manual annotation would help to understand how high-quality annotation is ensured.***
>
> **A1:**
> For **temporal consistency**, we simultaneously check three consecutive frames to ensure consistent annotations. To ensure **topological correctness**, we show the perspective view (PV) and bird's-eye-view (BEV) simultaneously to enable cross-validation between two different views. As for **attributes of traffic elements**, multiple stages of validation are processed so that a single frame is validated by several different inspectors.
>
> > ***Q2: The annotation range (+/-50 meters along the x-axis) might be too short for real-time applications.***
>
> **A2:**
> Agreed. In fact, we have extended the range compared to existing HD map learning works, whose range is +/-30m. As most of the participants in the hosted [Challenge](https://opendrivelab.com/AD23Challenge.html#openlane_topology) perform relatively poor in the long-range perception, we encourage the community to improve the model performance on current range and will add further long-range annotation as an important future development.
>
> > ***Q3: The data is collected mainly in the US. It's not clear whether models trained on OpenLane-V2 will be able to generalize to different countries.***
>
> **A3:**
> Agreed. As mentioned in the limitation that the proposed dataset is built on top of existing datasets, the diversity and scale of the dataset are limited. Nevertheless, the two subsets in this dataset have provided a platform for further explorations on the domain generalization problem. We will keep working on the opportunity to improve our data scale.
>
> > ***Q4: Traffic elements are annotated in 2D only. The missing 3D information might help in the centerline-traffic element association for the models.***
>
> **A4:**
> Yes, the 3D location of traffic elements may help. However, the range of sensors that provide 3D location information is limited in the height dimension, and there are traffic elements whose 3D location is unknown and hard to annotate. As a result, we only include 2D annotations for traffic elements.

---

> > ### Comment · Reviewer_bqmb · 2023-08-29
> >
> > Thank you to the authors for the detailed answers. My concerns have been resolved. I will keep my rating.

---

### Official Review · Reviewer_79Ej · 2023-07-21
**Interesting paper, some details are missing.**

**Rating:** 6
**Confidence:** 3
**Correctness:** The claims are correct
**Clarity:** The paper is clear and easy to follow.

**Strengths:**

The paper tackles an important problem and it is very easy to follow.

**Additional Feedback:**

See above.

**Documentation:**

Yes, they used the Datasheets for Datasets as reference,

**Ethics:**

No ethical concerns.

**Limitations:**

The limitations were discussed in the paper.

**Opportunities For Improvement:**

There are a few points on which the paper can improve:
1. The proposed baselines perform very poorly according to the TOP_ll metric
2. The authors write that the collected scenes come are from different places worldwide. However, they do not specify from where. Additionally, it would be interesting to see the percentages of scenes coming from each country.
3. Connected to the point above, I noticed that the average number of centrelines per frame is really high. This makes me think that these scenes are collected from very specific places (e.g., US). It would be interesting if the authors could explain why there are such a high number of centrelines per frame.
4. For the subset_A the authors write that the front-view images are transposed to $1550 \times 2048$. Why?
5. Also, the two subsets have different numbers of cameras, does this pose any problems?
6. Why do you label the attributes of invisible or irrelevant elements as unknown? How can you have an invisible element? What is the advantage with respect to not detecting them at all?
7. Can you provide an actual formal for DET_l? Right now the metric is not defined uniquely, making the results not reproducible.

**Relation To Prior Work:**

Yes, it is only missing the citation for the ROAD dataset [1] when it talks about scene understanding.

[1] Singh G, Akrigg S, Maio MD, Fontana V, Alitappeh RJ, Khan S, Saha S, Jeddisaravi K, Yousefi F, Culley J, Nicholson T, Omokeowa J, Grazioso S, Bradley A, Gironimo GD, Cuzzolin F. ROAD: The Road Event Awareness Dataset for Autonomous Driving. IEEE Trans Pattern Anal Mach Intell. 2023 Jan;45(1):1036-1054. doi: 10.1109/TPAMI.2022.3150906. Epub 2022 Dec 5. PMID: 35157577.

**Summary And Contributions:**

The paper proposes a novel dataset, called OpenLane-V2 for simultaneous 3D lane and traffic entities detection. The paper is built upon the two dataset: Argoverse and nuScenes. Each scene has been annotated with the 3D projection of the lanes together with the position of the traffic entities that are relevant to decide which is the set of actions available for the ego vehicle (i.e., the set of actions that it can do).

---

> ### Author Response · Authors · 2023-08-12
> **Author response to Reviewer 79Ej**
>
> Dear Reviewer 79Ej,
>
> Thank you for your comments. We address your concerns on the weaknesses below.
>
> > ***Q1: The proposed baselines perform very poorly on the $\text{TOP}_{ll}$ metric.***
>
> **A1:**
> The task of topology reasoning is challenging, as the ground truth and predictions are required to be matched within a small distance threshold. In the [Challenge](https://opendrivelab.com/AD23Challenge.html#openlane_topology) we hosted, most of the participating teams built their models based on the provided baseline and made an improvement. The codebase is helpful for researchers to get started with the proposed task.
>
> > ***Q2: The authors write that the collected scenes come from different places worldwide. It would be interesting to see the percentages of scenes coming from each country.***
>
> **A2:**
> As provided in `Line 136` in the updated version, statistics on data distribution are as follows:
> *The subset-A is collected in the USA, while the the subset-B is collected from two countries: USA (55.0%) and Singapore (45.0%).*
>
> > ***Q3: The average number of centrelines per frame is really high. Why there are such a high number of centrelines per frame?***
>
> **A3:**
> For most scenes, there exists an intersection where the number of centerlines is high. As shown in the example of a simple intersection in `Figure 5 in Supplementary`, 8 centerlines are required for two lanes. For intersections with more complex structures, the number of centerlines would grow larger.
>
> > ***Q4: Why the front-view images are transposed to 1550×2048 for the subset-A. Also, the two subsets have different numbers of cameras, does this pose any problems?***
>
> **A4:**
> The front center camera is transposed to improve the vertical field of view so that the image can capture traffic elements in higher positions.
>
> The difference in the sensor setting would lead to the domain gap, which is a widely existing issue for deep learning.
> To the best of our knowledge, currently few works focus on this domain. We hope that this characteristic of our dataset could facilitate relevant research.
>
> > ***Q5: The paper labels the attributes of invisible elements as unknown. However, it is unclear how an element can be considered invisible, and what advantage this labeling approach provides over not detecting these elements at all.***
>
> **A5:**
> Invisible traffic elements represent those too small or too vague to recognize the semantic category, but their existence could be spotted.
> Detecting such elements would build a relationship between centerlines and traffic elements, and suggest that there might be an intersection, for instance.
> We have adjusted the content for clarity in the updated version in `Line 173`.
>
> > ***Q6: A formal definition or actual example for DET_l to ensure that the metric is reproducible, even if it is jointly defined.***
>
> **A6:**
> The DET_l score adopts mAP in the field of object detection. We use Fréchet distance as the similarity measure to decide a match between ground truth and predictions. The APs are calculated under different match thresholds and then averaged as the DET_l score. In Equation 4 in the paper, AP is defined as the area under the precision-recall curve, which is represented as $\int_0^1 p(r) \text{d}r$, where $p$ and $r$ denote precision and recall respectively. The manuscript is updated in `Line 215`.
>
> > ***Relation To Prior Work: Missing the citation for the ROAD dataset when it talks about scene understanding.***
>
> **A:**
> Thank you for bringing us the related work. We have added the mentioned work in the revised version in `Line 117`.

---

### Official Review · Reviewer_wxkw · 2023-07-21
**OpenLane-V2: A Topology Reasoning Benchmark for Scene Understanding in Autonomous Driving**

**Rating:** 4
**Confidence:** 5
**Correctness:** Yes
**Clarity:** Yes

**Strengths:**

* OpenLane-V2 dataset  is an effort to bridge the gap between various disjointed datasets in autonomous driving, offering a comprehensive understanding of driving scenes by integrating both lanelines and traffic signals.
* OpenLane-V2 is also the first dataset that works on topology reasoning which is useful for autonomous driving system


**Additional Feedback:**

I would be happy to increase my score if the author can address my concerns listed in the 'Opportunities For Improvement' section

**Documentation:**

Yes

**Ethics:**

None.

**Limitations:**

The authors should provide insights into the diversity of conditions under which data was collected. This includes various weather conditions, times of the day, geographical locations, and traffic scenarios. A diverse dataset is crucial for training robust autonomous systems.

**Opportunities For Improvement:**

* Even though VectorMapNet and MapTR are included as baselines, HD Map learning as a growing research topic should be compared to in this paper. HD map learning is also a benchmark for Scene Understanding in autonomous driving and it's also based on nuScenes and argoVerse. I think the additional value of OpenLaneV2 based on HD Map Learning should be explained in this paper.
* The novelty of OpenLaneV2 is incremental rather than groundbreaking, especially if we compare it with OpenLane or HD Map Learning.
* The traffic element dataset only has 13 classes which is not very useful in real-life. As in table 1, usually more classes are included in a traffic element dataset.
* I don't understand why this dataset tries to promote centerlines instead of lanelines. The author wrote 'lanelines are divided arbitrarily' but the centerlines in this dataset are extracted from lanelines. Also, lanelines are visible markings that have much less ambiguity than centerlines. The worldwide road systems are designed based on lanelines.
* I am not sure 'many to many' mappings between traffic elements and lanes are addressed in this dataset.



**Relation To Prior Work:**

Again, i think HD map learning is a very similar benchmark that needs to be included and compared to in this dataset paper.

**Summary And Contributions:**

While many datasets focus on individual aspects like lanelines or traffic signals, there's a gap in comprehensively integrating these elements. Existing 2D datasets fall short in real-world scenarios, especially when translating into a bird’s-eye-view or at unmarked crossroads. To bridge this gap, the paper introduces the OpenLane-V2 dataset, aiming to provide a holistic understanding of driving scenes. This dataset captures both scene structure and topological relationships, with metrics (OpenLane-V2 Score) assessing detection and reasoning. Unlike predecessors, OpenLane-V2 offers 3D lane annotations, directed lane centerlines, and annotates relationships between traffic elements and lanes. Comprising images from various global cities, it contains 2.1M instance-level annotations and 1.9M topology relationships. Additionally, the authors provide a development kit and plug-ins for deep learning frameworks, ensuring easy accessibility and fair model comparisons.

---

> ### Author Response · Authors · 2023-08-12
> **Author response to Reviewer wxkw**
>
> Dear Reviewer wxkw,
>
> Thank you for your helpful review. We address your concerns below.
>
> > ***Q1: The additional value of OpenLaneV2 based on HD Map Learning should be explained, especially when comparing it with OpenLane or HD Map Learning.***
>
> **A1:**
> We agree with the reviewer that HD Mapping is a growing research topic. We address this problem and add the comparisons in the updated manuscript in `Line 34` and `Line 52`:
> 1. OpenLane focuses on 3D laneline detection. Compared to it, we provide additional annotations on **traffic elements and topology relationships**.
> 2. HD map learning task formulates visible lanelines as rasterized or vectorized representation. We provide an instance-level representation of lanes, enabling topology relationships of different entities.
>
> > ***Q2: The traffic element dataset only has 13 classes which are not very useful in real-life.***
>
> **A2:**
> Thanks and agreed.
> We pay more attention to **topology reasoning** than the semantic meaning of traffic elements (red, green, *etc.*).
> We will improve the variety and regard the task of traffic element recognition as an open-vocabulary problem in the future.
>
> > ***Q3: Why this dataset tries to promote centerlines instead of lanelines? The author wrote 'lanelines are divided arbitrarily', but the centerlines extracted from lanelines are more ambiguous than the visible lanelines.***
>
> **A3:**
> As mentioned in `Line 36-41`, we reckon that centerline is an important representation for downstream tasks. The visible lanelines cannot be directly used by downstream tasks without laborious postprocessing. As depicted in `Figure 2 in Supplementary`, lanelines cannot represent lanes, as a laneline is shared by two lanes. Besides, compared to lanelines, properties can be attached to centerlines, such as the lane direction.
>
> With all due respect, centerlines do not lead to ambiguity. As shown in the left part of `Figure 3 in Supplementary`, a single laneline is split into several segments without visually apparent hints in the HD map. While in the right part of `Figure 3 in Supplementary`, for constructing centerlines, we have a process to merge centerlines into a more complete format. As a result, centerlines are only divided in some particular positions with apparent indications, such as splitting and merging of lanes.
>
> We have rephrased it accordingly in the updated version in `Line 146`.
>
> > ***Q4:'many to many' mappings between traffic elements and lanes are not addressed.***
>
> **A4:**
> Indeed, 'many to many' mappings are provided. For instance, the red light is associated with multiple centerlines, as shown in `Figure 4 in Supplementary`.
>
> > ***Limitations: The authors should provide insights into the diversity of conditions under which data was collected.***
>
> **A:**
> Thanks. We provide statistics on data distribution in `Line 136` in the updated version, and they are as follows:
> *Concerning geographical locations, comprises scenes from six cities: Austin (3.1%), Detroit (11.7%), Miami (35.4%), Pittsburgh (35.0%), Palo Alto (2.2%), and Washington D.C. (12.6%), while the the subset-B is collected from two cities: Boston (55.0%) and Singapore (45.0%). The subset-A includes 3.0% night scenes and 1.1% rain scenes, while the subset-B includes 11.7% night scenes and 17.4% rain scenes.*

---

### Official Review · Reviewer_hUvy · 2023-07-22
**A large-scale dataset based on Argoverse and nuScenes for joint perception of lane and traffic signals.**

**Rating:** 6
**Confidence:** 4
**Correctness:** Yes.
**Clarity:** Yes.

**Strengths:**

1. Comprehensive dataset: The OpenLane-V2 dataset unifies lanes, traffic elements, and topology relationships, providing a more realistic and holistic map representation of driving scenarios.

2. 3D lane annotations: The authors introduce 3D lane centerlines and their connectivity in a bird's-eye-view space, reflecting real-world driving properties more accurately and overcoming the limitations of 2D lane annotations.

3. Holistic metric: The OpenLane-V2 Score (OLS) combines perception and reasoning components, offering a comprehensive evaluation of model performance on the proposed benchmark.

4. Extensive annotations: The dataset contains a substantial number of instance-level annotations and positive topology relationships, making it a valuable resource for training and evaluating autonomous driving models.

5. Community involvement: The provision of a development kit, plug-ins for deep learning frameworks, and a test server with a leaderboard demonstrates the authors' commitment to community involvement and fair comparisons.

**Additional Feedback:**

I will consider raising my rating if the authors address my concerns.

**Documentation:**

Yes.

**Ethics:**

No.

**Limitations:**

Yes.

**Opportunities For Improvement:**

The paper's title, "A topology reasoning benchmark for scene understanding," appears to be overly broad compared to the actual contribution presented in the work. The paper indeed proposes a major improvement in the field of 3D HD mapping by unifying lane perception, traffic signals, and their association, distinguishing it from previous 2D HD mapping approaches like MapTr and VectorMapNet. However, it is crucial to recognize that the proposed benchmark focuses on specific aspects of scene understanding and does not fully address the complexities of dynamic objects, static environments, and the topology relationship between various complex traffic elements such as cars, pedestrians, cyclists, and road elements.

To avoid overclaiming and accurately represent the paper's contribution, a more appropriate title could be: "A Unified 3D HD Mapping Benchmark for Lane Perception and Traffic Signal Association." This revised title highlights the specific areas of improvement and aligns more closely with the content of the paper.

I am also confused about the TOP score. The authors are suggested to give some intuitions behind this design and include several examples to help readers better understand it.

While the paper convincingly motivates the need for topology relationships between lanes and traffic signals, specific examples of how these relationships can benefit downstream tasks would strengthen the paper's argument for the dataset's practical value.

**Relation To Prior Work:**

Yes.

**Summary And Contributions:**

The paper presents a novel large-scale dataset, OpenLane-V2, that unifies the perception of lanes, traffic elements, and their relationships, thereby enhancing decision-making capabilities in complex driving environments. The dataset introduces 3D lane annotations, allowing for a more accurate representation of real-world driving scenarios. Additionally, the authors propose a comprehensive evaluation metric, the OpenLane-V2 Score (OLS), which assesses model performance on the dataset. To support the research community, the paper includes a development kit and plans for ongoing maintenance of deep learning framework support and a test server. Overall, the OpenLane-V2 dataset and associated resources offer a valuable contribution to the field of autonomous driving research.

---

> ### Author Response · Authors · 2023-08-12
> **Author response to Reviewer hUvy**
>
> Dear Reviewer hUvy,
>
> Thanks. We address your concerns below.
>
> > ***Q1: The paper's title appears to be overly broad compared to the actual contribution presented in the work. The proposed benchmark focuses on specific aspects of scene understanding and does not fully address the complexities of dynamic objects, static environments, and the topology relationship between various complex traffic elements. A more appropriate title could be: "A Unified 3D HD Mapping Benchmark for Lane Perception and Traffic Signal Association.***
>
> **A1:**
> We agree that our work does not include dynamic objects.
> As mentioned in `Line 6` and `Line 21` in the manuscript, 'driving scene or traffic scene' refers to the background information.
> We are pleased to discuss with Reviewer about the title. As requested, we propose to change the title to "OpenLane-V2: A Unified 3D HD Mapping Benchmark for Topology Reasoning".
>
> > ***Q2: Confusion about the TOP score. Some intuitions behind this design and several examples should be provided to help readers better understand it.***
>
> **A2:** The TOP score evaluates the performance on topology reasoning, which can be seen as a graph learning problem, as described in `Line 234-244`. In the area of classical graph learning, mean Average Precision(mAP) is a commonly-used metric to measure the link prediction for each vertex. The TOP score adopts mAP for vertex-level evaluation and then takes the average of mAP results over all vertices to reflect the model’s overall topological reasoning ability on the dataset. We have added more explanations for readability accordingly in the updated manuscript (`Line 245` and `Line 251`).
>
> For instance, given that a graph has eight vertices, vertex 1 of which has adjacent neighbors {2, 3, 5, 8} and the predicted result is {4, 3, 5, 6, 7}, we calculate the mAP score of vertex 1 as follows:
>
> $$mAP(1) = \frac{FP(4) + TP(3) + TP(5) + FP(6) + FP(7)}{|\{2, 3 ,5 ,8\}|} = \frac{0 \times 0 + 1 \times \frac{1}{2} + 1 \times \frac{2}{3} + 0 \times \frac{2}{4} + 0 \times \frac{2}{5}}{4} = \frac{7}{24}$$
>
> Here, for each multiplication term in the numerator, the first multiplier indicates whether the predicted connection is correct or not, and the second one is the precision of the predicted neighbors, as they are sorted according to their confidence.
> Supposed that the mAPs of other vertices are 0, the TOP score is averaged over all vertices:
> $$TOP = \frac{1}{8} \sum_{i=1}^{8} mAP(i) = \frac{\frac{7}{24} + 0 + 0 + 0 + 0 + 0 + 0 + 0}{8} = \frac{7}{192} \approx 0.036$$
>
> We have added the example to `Line 93 in Supplementary`.
>
> > ***Q3: Specific examples of how the relationships between lanes and traffic signals can benefit downstream tasks would strengthen the paper's argument for the dataset's practical value.***
>
> **A3:** As illustrated in `Figure 1 in Supplementary`: (a) two vehicles are at the same speed; (b) the left vehicle needs to slow down as the associated traffic light turns red, while the right one does not. The topology relationships enable the vehicle to recognize the signal it should follow.
>
> In the recent [nuPlan](https://www.nuscenes.org/nuplan) benchmark for closed-loop planning, the traffic status of a particular lane (centerline specifically) is directly provided as **extra guidance** for autonomous vehicles and serves as model input. In our dataset, real-time traffic information is assigned to specific lanes via topology relationships, aligning with human agents' instincts. We have adjusted the content accordingly in the updated version in `Line 48`.

---

> > ### Comment · Reviewer_hUvy · 2023-08-30
> >
> > Thank authors for addressing my concerns. Overall, I think this work is a nice contribution to the current autonomous driving community. I hope the authors can further polish the title and make reasonable claims in the final version. I will raise my score to 6.

---

### Author Response · Authors · 2023-08-12
**General Author Response for Rebuttal**

Dear Reviewers and AC(s),

We appreciate your careful reviews and detailed comments.
We have added some details to address concerns from reviewers in the rebuttal.
Please see each response below, and the updated PDF manuscript, **where the revised paragraph is marked in `Blue`.**


Ever since the release of Openlane-V2, we have received many issues / pull requests / emails from the community. We have also hosted a [Challenge](https://opendrivelab.com/AD23Challenge.html#openlane_topology) at CVPR 2023, attracting a large group of audience from different regions and institutions.
We will continue to maintain the test server as a regular track year around for incoming submissions, as our commitment to the research community.

Thanks,
OpenLane-V2 authors.

---

> ### Author Response · Authors · 2023-08-28
> **Discussion ending in One day**
>
> Hi Reviewers,
>
> Not sure if PCs / ACs have sent the reminder. The disscussion between authors/reviewers are soon due in one day.
>
> We express our sincere gratitude for valuable reviews, and we have taken into account each comment provided.
>
> If there are any remaining issues or concerns, please inform us! Thank you for the time being involved.
>
> Best,
> Authors

---

### Decision · Program_Chairs · 2023-09-22

**Decision:**

Accept (Poster)

**Comment:**

The paper presents a dataset for autonomous driving called OpenLane-V2 (extending the previous OpenLane) which augments lane annotations with traffic elements and topology relationship annotations. 2K road scenes have been annotated. 3 tasks and multiple competent baseline methods have been benchmarked on the proposed dataset. While there are some concerns from the reviewers regarding the clarify of the title, the difference compared to HD Map Learning, and many details that require further clarification, most of which have been well addressed during the rebuttal. The reviewers and AC agree mostly on recognizing the contributions of this paper on the dataset and annotation contribution, and its significance and importance to the field of study. Therefore, the AC recommends accepting this paper.